

# Effects of habitat fragmentation and human disturbance on the population dynamics of the Yunnan snub-nosed monkey from 1994 to 2016

Xumao Zhao[1,2], Baoping Ren[1], Dayong Li[3], Zuofu Xiang[4], Paul A. Garber[5] and Ming Li[1,6]

[1] CAS Key Laboratory of Animal Ecology and Conservation Biology, Institute of Zoology, Chinese Academy of Sciences, Beijing, China
[2] University of the Chinese Academy of Sciences, Beijing, China
[3] Key Laboratory of Southwest China Wildlife Resources Conservation (Ministry of Education), China West Normal University, Nancong, China
[4] College of Life Science and Technology, Central South University of Forestry and Technology, Changsha, China
[5] Department of Anthropology and Program in Ecology and Evolutionary Biology, University of Illinois, Urbana, IL, USA
[6] Center for Excellence in Animal Evolution and Genetics, Chinese Academy of Sciences, Kunming, China

Corresponding author
Ming Li, lim@ioz.ac.cn

## ABSTRACT

In this study, we integrate data from field investigations, spatial analysis, genetic analysis, and Generalized Linear Models (GLMs) to evaluate the effects of habitat fragmentation on the population dynamics, genetic diversity, and range shifts in the endangered Yunnan snub-nosed monkey (*Rhinopithecus bieti*). The results indicate that from 1994 to 2016, *R. bieti* population size increased from less than 2,000 to approximately 3,000 individuals. A primary factor promoting population recovery was the establishment of protected nature reserves. We also found that subpopulation growth rates were uneven, with the groups in some areas, and the formation of new groups. Both the fragmentation index, defined as the ratio of the number of forest patches to the total area of forest patches (e.g., increased fragmentation), and increasing human population size had a negative effect on population growth in *R. bieti*. We recommend that government conservation plans prioritize the protection of particular *R. bieti* populations, such as the Baimei and Jisichang populations, which have uncommon haplotypes. In addition, effective conservation strategies need to include an expansion of migration corridors to enable individuals from larger populations such as Guyoulong (Guilong) to serve as a source population to increase the genetic diversity of smaller *R. bieti* subpopulations. We argue that policies designed to protect endangered primates should not focus solely on total population size but also need to determine the amount of genetic diversity present across different subpopulations and use this information as a measure of the effectiveness of current conservation policies and the basis for new conservation policies.

## INTRODUCTION

Globally, anthropogenic activities resulting in deforestation, habitat loss, the extraction of natural resources, the introduction of invasive species, agricultural expansion, and climate change represent a major driver of the unprecedented decline in biodiversity (*Millennium Ecosystem Assessment, 2006*; *Magurran & Dornelas, 2010*; *Newbold et al., 2015*). Quantifying the effects of individual factors and developing solutions to protect environments and endangered is a key scientific challenge (*Mace et al., 2010*; *Magurran & Dornelas, 2010*). Anthropogenic activities have altered as much as 50% of terrestrial land cover, and land-use patterns, such as the expansion of croplands for industrial agriculture and pastures for grazing cattle and other domesticated animals, has led to a global reduction in the number of species (*Millennium Ecosystem Assessment, 2006*; *McGill, 2015*). This has led scientists to refer to the current period as the Anthropocene mass extinction event (*Ceballos, Ehrlich & Dirzo, 2017*), with species extinction rates estimated to be 100–1,000 times greater than during past evolutionary periods (*Raup, 1995*; *Pimm et al., 1995*). Protecting animal and plant survivorship ad biodiversity requires monitoring and managing changes in species population size and genetic diversity (*Dornelas et al., 2014*).

An important form of habitat degradation is fragmentation, which transforms large tracts of continuous habitat into smaller and spatially distinct patches immersed within a dissimilar matrix (*Didham, Kapos & Ewers, 2012*; *Wilson et al., 2016*). Fragmentation can create detrimental edge affects along the boundaries of habitat patches, alter ecological conditions resulting in the decline of native species, restrict animal movement and gene flow, and sever landscape connectivity (*Crooks & Sanjayan, 2006*) leading to local population extinction in response to insufficient viable "core" habitat (*Ewers & Didham, 2007*). Habitat fragmentation has continued at an alarming rate, and has impaired key ecosystem functions by decreasing biomass, limiting seed dispersal, shifting predator-prey relationships, and altering nutrient cycles (*Haddad et al., 2015*).

Several species of nonhuman primates are particularly affected by habitat fragmentation due to their dependence on intact and biodiverse forested landscapes to obtain a nutritionally balanced diet (*Arroyo-Rodríguez & Fahrig, 2014*). Currently, anthropogenic habitat modifications have resulted in more than half of the world's primate species listed as Vulnerable, Endangered, or Critically Endangered (*Estrada et al., 2017*, *2018*). Primate populations inhabiting small and isolated forest fragments are especially vulnerable to extinction (*Benchimol & Peres, 2013*). However, given evidence of behavioral plasticity, the benefits of social learning, and the ability to exploit a broad range of different food types, many species of primates are reported to survive, at least over periods of several years, in impacted landscapes by modifying aspects of their activity budget, group size and composition, ranging patterns, and diet (*Onderdonk & Chapman, 2000*; *Wong & Sicotte, 2007*; *Boyle & Smith, 2010*).

The endangered Yunnan snub-nosed monkey (*Rhinopithecus bieti*), a large-bodied species of nonhuman primate, that lives in social groups of several 100 individuals (*Kirkpatrick et al., 1998*). A survey in the early 1990s indicated that this species was

confined to a narrow region between the Yangtze and Mekong rivers (98°37′–99°41′E, 26°14′–29°20′N). Their population size was estimated to be less than 2,000 individuals, distributed across 19 distinct social groups (*Long et al., 1994*). A study by *Zhao et al. (2018)* integrating data on evolutionary genetics and biogeographical information concluded that both the historical distribution (the past 2,000 years) and current population structure of the *R. bieti* appears to have been directly impacted by human activities, principally agricultural expansion resulting in severe habitat fragmentation of the Tibetan Plateau and by hunting (*Liu et al., 2009*).

Recent studies modeling habitat change and agricultural expansion across *R. bieti's* range, predict an increase in forest fragmentation and a decrease in habitat quality resulting in range contraction over the next 25–75 years (*Xiao et al., 2003*; *Wong et al., 2013*; *Li et al., 2018*). However, some of these studies were based on data collected in the early 1990s and therefore may not accurately represent the current demographic and ecological challenges and conservation pressures faced by *R. bieti*. The main purpose of our study is to: (1) investigate the current population size and distribution of the *R. bieti*, (2) conduct a landscape and spatial analysis of their distribution area, and (3) examine how present day habitat fragmentation influences *R. bieti* population size, genetic diversity, and changes in geographical distribution.

## METHODS

### Study area and data collection

This field study was approved by State Forestry Administration of China. The study area is a narrow region of 17,000 km$^2$ in the northwest of Yunnan Province and the Tibet Autonomous Region. It is bounded by the Mekong River to the west and the Yangtze River to the east (*Long et al., 1994*). The elevation of the study area varies from 1,300 to 5,400 m. Our study was conducted across all current distributional areas of *R. bieti* in Yunnan Province and Tibet. Each of the remaining wild groups was surveyed from January to November 2013 and April to September 2016. In order to obtain the most accurate information for this species, we conducted detailed surveys within its main distribution in three national nature reserves (Bamaxueshan, Honglaxueshan and Tianchi), one provincial nature reserve (Yunling), and two sites that are outside of protected reserves (Jinsichang and Bamei) (Fig. 1).

Based on information of group location, a survey was conducted and the number of monkeys per group was censused. Specific locations of *R. bieti* groups were obtained from local forest rangers and officials who regularly patrol the reserves. *R. bieti* were counted directly using auxiliary telescopes to assist in observations (*Wu, Zhong & Wu, 1988*). Each monkey group was observed for 14–21 days, counting all individuals in the group (*Wu, Zhong & Wu, 1988*). The counting of monkeys in each group was based on the following method. Travel in *R. bieti* occurs both on the ground and in trees. However, when crossing open areas, the monkeys often walk slowly in a single or nearly single. A total of 17 groups were censused. *R. bieti* live in groups called multilevel societies (MLS), which are composed of several harem social and reproductive units (*Qi et al., 2014*). Each harem contains a single adult male, several adult females, and offspring.

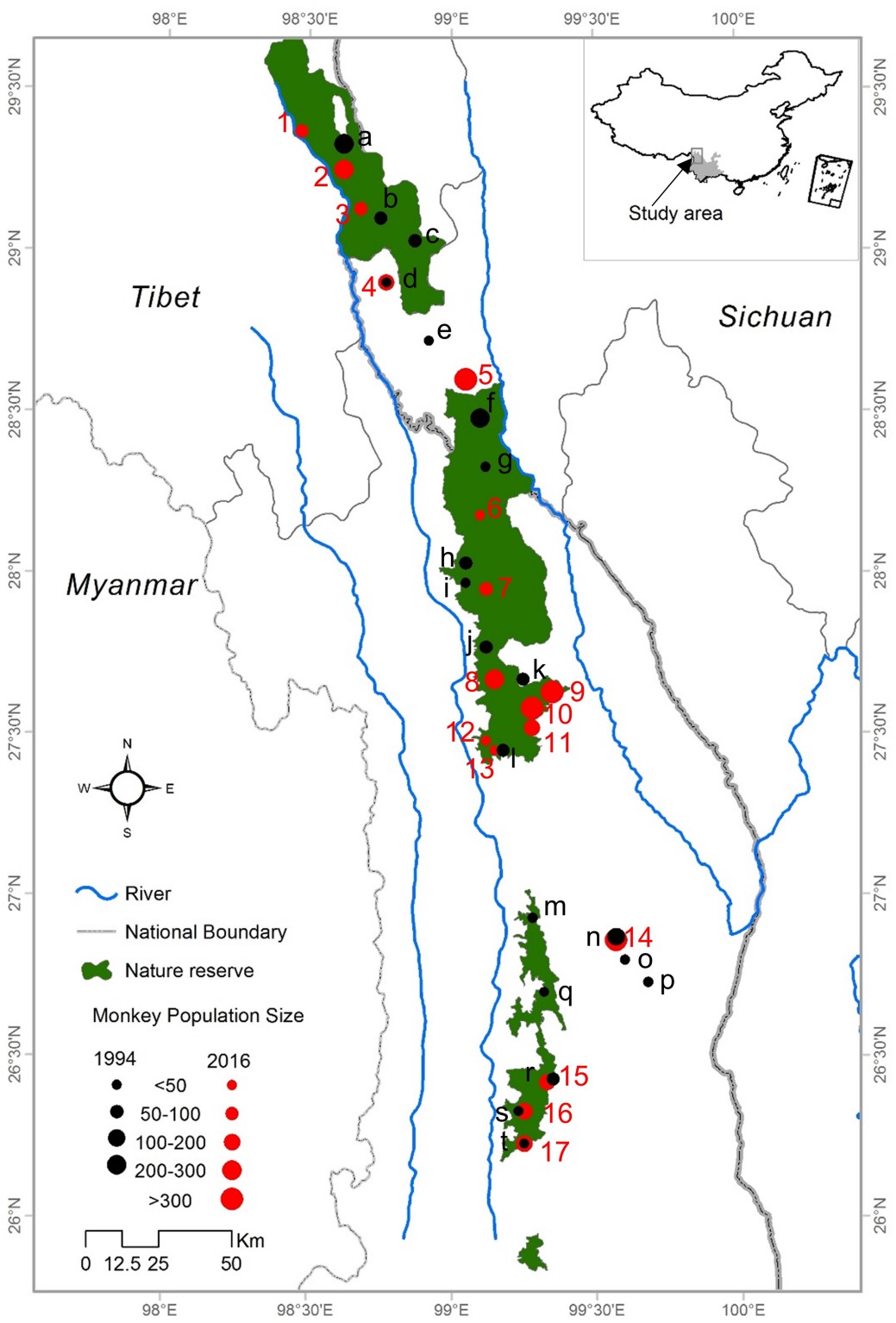

**Figure 1 The study area and locations of the bands of *R. bieti* from 1994 to 2016.**

These harems travel, feed, and range together throughout the day and form a large band. Bands are followed by one or more all male units (AMU) that contain juvenile, sub-adult, and adult bachelor males. Because each band or MLS has been tracked by patrol officers for several years, we were able to obtain an accurate count of its size. This was accomplished by first counting the number of harems in each band and then estimating the number of monkeys per harem and per AMU to obtain a final total. In order to increase census accuracy, at least three researchers participated in each survey and counted/estimated the number of harems and harem sizes.

To monitor changes over time in *R. bieti* demography and distribution, we obtained information on *R. bieti* population size and location in 1990 from *Long et al. (1994)* and compared it with our current data. To assess relationships among suitable habit, population size, and gene flow, we conducted an analysis of genetic diversity among groups. The genetic diversity for each group was calculated using FSTAT 3.2 (*Goudet, 1995*) based on microsatellite dates from the analyses of 157 individuals' samples (blood, muscle and faecal) in 2007 (*Liu et al., 2007*, *2009*). Genetic information was obtained from 11 of the 17 known groups of *R. bieti*.

## Landscape data and spatial analysis

We used a land cover map ($30'' \times 30''$ resolution) derived from an assemblage of 13 SPOT5 images across the known distribution of *R. bieti* for 1990 and 2016. This covered an area of 17,000 km$^2$ including the entire *R. bieti* range. Vegetation types suitable for *R. bieti* were defined as dark-coniferous forest dominated by species such as *Abies georgei* and mixed coniferous and broadleaf forest (*Kirkpatrick et al., 1998*; *Xiao et al., 2003*; *Li et al., 2015*). We then verified our habitat classification accuracy using data from the Conservation Information Centre of The Nature Conservancy of China. Fragmentation metrics for *R. bieti* were calculated using FRAGSTATS 4 (*McGarigal & Marks, 1994*). Patch density (PD, the number of patches divided by total landscape area) was used as indices of fragmentation, with low PD indicating a more connected landscape and high PD indicating a more fragmented landscape (*Olsoy et al., 2016*). A fragment was considered vegetatively suitable to sustain a population of *R. bieti* if it included dark-coniferous forest and mixed coniferous and broadleaf forest. Given the potential effect that habitat fragmentation can have on whether an area is suitable for *R. bieti*. Supplementary information of forested land, cropland, pasture land, and land reclamation (defined as primary forest converted into a secondary forest, pasture and cropland), as well as information on human population density, was extracted from the History Database of the Global Environment (the pixel of size of these data were $0.25 \times 0.25$ resolution; HYDE3.1: *Goldewijk et al., 2011*; *He, Li & Zhang, 2015*; *Li, He & Zhang, 2016*). We extracted values of human population density, cropland, and pasture (HYDE3.2; *Goldewijk, 2016*), all indicators of habitat fragmentation, between the years 1990 and 2016 with the function "extract" in R (*R Core Team, 2017*). We used Spearman correlations to check for simple linear relationships between the growth rate of the *R. biet* population and all environmental and landscape variables (fragmentation index, population density, area of cropland, area of pasture, density of households, and density of roads). Then, we

applied multiple linear regression analyses to determine the partial and interactive effects of factors affecting *R. bieti* population size. Finally, we applied GLMs to determine the factors affecting *R. bieti* population size.

## RESULTS

Based on a census of 17 *R. bieti* subpopulation size isolated from each other and troop size ranged from 40 to 450 individuals. The mean number of harems per band was 12 ± 3.5. Across all subpopulations, the total number of *R. bieti* in 2016 was estimated at approximately 3,000 individuals. We found that population growth had varied significantly by region. Current population estimates in the central and southwestern zones of the species' range (Fig. 1) (Baimaxueshan Nature Reserve and Yunling Nature Reserve) were 147% higher compared to values reported in 1994 (including the formation of three new groups: Gehuaqing, Baijixun (Yongan), Shikuadi). In contrast, size estimates for the southeastern and northern (Honglaxueshan Nature Reserve) areas increased by only 24% and 21%, respectively, from earlier estimates. This increase in population size was the result of an increase in the size of bands rather than the number of *R. bieti* bands. We found that six groups with populations sizes of less than 100 individuals in 1994 disappeared by 2016. These included those in Bajia, Adong, Houziqin, Heishan, Dapingzi, and Moziping. The population sizes of three groups (Cikatong, Milaka, and Akou (Anyi)) were found to decrease by 20–50% between 1994 and 2016 (Table 1). Four new groups were identified in our census, Zhina, Gehuaqing, Baijixun (Yongan), and Shikuadi. These bands inhabit the northern and southern regions of the species distribution and ranged in size of from 40 to 450 individuals (Table 1).

Our analysis of habitat fragmentation indicates that the southern populations of *R. bieti* (Tianchi nature reserve, Yunling nature reserve and Jinsichang, fragmentation index: 0.99), live in more fragmented habitats and are characterized by smaller band size compared to the Central ($N = 9$) and northern ($N = 3$) populations (Baimaxueshan nature reserve and Honglaxueshan nature reserve, fragmentation index: 0.59). In general, the GLMs indicated that the fragmentation index, human population size, and the area within a group's range devoted to pasture was negatively correlated with *R. bieti* population size. In this case, after model selection based on AIC, only a few variables remained (population size = 0.85–3.09 fragmentation index −2.39 population density −0.49 pasture, $R^2 = 0.52$, $P < 0.05$).

## DISCUSSION

The results of our census indicate that the construction of protected nature reserves (87% of the *R. bieti* live in protected areas) has effectively limited human disturbance and reduced hunting, leading to a population increase from less than 2,000 to almost 3,000 individuals over the past 25 years. During the 1980s, populations of *R. bieti* faced a set of severe anthropogenic challenges. For example, approximately 200,000 m$^3$ of commercial logs were removed annually from their range, an entire band was killed by poachers (*Long et al., 1994*; *Ding, Yang & Liu, 2003*), and the area of suitable habitat decreased by 31% from 1958 to 1997 as large tracts of forests were converted into pastures

**Table 1 Changes in the location of natural bands of *R. bieti* between 1994 to 2016.**

| Site | Latitude | Longitude | 1994 | 2016 | Change group | Intraspecific genetic diversity | |
|---|---|---|---|---|---|---|---|
| | | | | | | Unbiased Hz | Obs Hz |
| Zhina | 29.37 | 98.47 | – | 80 | new | 0.54 | 0.50 |
| Xiaochangdu | 29.33 | 98.62 | >200 | 280 | unchanged | – | – |
| Milaka | 29.13 | 98.75 | <100 | 60 | unchanged | 0.58 | 0.64 |
| Bajia | 29.03 | 98.87 | <100 | – | disappeared | – | – |
| Bamei | 28.9 | 98.77 | <50 | 100 | unchanged | 0.59 | 0.54 |
| Adong | 28.72 | 98.92 | <50 | – | disappeared | – | – |
| Wuyapuya | 28.48 | 99.1 | >200 | 400 | unchanged | 0.63 | 0.64 |
| Cikatong | 28.03 | 99.05 | 50–100 | 50 | unchanged | 0.63 | 0.65 |
| Guyoulong(guilong) | 27.97 | 99.05 | <50 | 80 | unchanged | 0.70 | 0.69 |
| Shiba | 27.77 | 99.12 | 50–100 | 200 | Unchanged | – | – |
| Guomorong(Xiangguqibg) | 27.67 | 99.25 | 50–100 | 480 | unchanged | 0.65 | 0.63 |
| Akou(Anyi) | 27.45 | 99.18 | 50–100 | 40 | unchanged | 0.66 | 0.59 |
| Jinsichang | 26.87 | 99.57 | 100–150 | 310 | unchanged | 0.57 | 0.54 |
| Neidaqin(Fuhe) | 26.43 | 99.35 | 50–100 | 120 | unchanged | 0.66 | 0.71 |
| Lashashan | 26.33 | 99.23 | <50 | 130 | unchanged | – | – |
| Houziqin | 26.93 | 99.28 | <50 | – | disappeared | – | – |
| Longma | 26.23 | 99.25 | <50 | 140 | unchanged | 0.64 | 0.72 |
| Heishan | 26.7 | 99.32 | <50 | – | disappeared | – | – |
| Dapingzi | 26.73 | 99.68 | <50 | – | disappeared | – | – |
| Moziping | 26.8 | 99.6 | <50 | – | disappeared | – | – |
| Gehuaqing | 27.58 | 99.28 | – | 450 | new | – | – |
| Shikuadi | 27.52 | 99.28 | – | 120 | new | | |
| Pantiange | 27.47 | 99.15 | – | – | unchanged | – | – |
| Baijixun(Yongan) | 27.48 | 99.12 | – | 40 | new | – | – |

for cattle grazing (*Xiao et al., 2003*). In addition, in Deqin country alone, approximately 430 *R. bieti* were killed for meat, fur, and medicine from 1970 to 1979, and 68 pieces of *R. bieti* fur or bone were sold in local stores (*Bai, 1987*). With the establishment of the Baimaxueshan Nature Reserve, the Honglashan Nature Reserve, the Yunling Nature Reserve and the Tianchi Nature Reserve between 1983 and 2003, *R. bieti* is now protected and this has effectively prevented poaching, the collection of forest products, and livestock grazing within their range. Simultaneously the Chinese government has banned all guns beginning in 1990s, and this has dramatically reduced poaching. Each of these measures has served to protect *R. bieti* and to promote their population recovery over the past 25 years.

Despite population increases, the conservation status of *R. bieti* continues to be negatively affected by forest fragmentation and the presence of human settlements in and around their range. Our results indicate that *R. bieti* population growth rates varied across its distribution with both increased habitat fragmentation and increased human population density negatively affecting *R. bieti* population growth rates.

This was most evident in the southern region (Yunling and Tianchi Nature Reserves), where suitable habitats are highly fragmented and *R. bieti* population growth rates are low. The situation faced by *R. bieti* of fragmentation-induced reduction in habitat quality and resource availability also has adversely affected the long-term viability of other primate species (*Wahungu et al., 2005*; *Arroyo-Rodríguez & Fahrig, 2014*) and this has contributed to a primate extinction crisis worldwide (*Estrada et al., 2017*, *2018*; *Li et al., 2018*).

In the case of *R. bieti*, forest fragmentation spatially isolates subpopulations and impeding gene flow. Moreover, the conversion of natural habitat to agricultural fields and pasture land leads to the local extinction of small populations, as in the case of the Adong, Houziqin, Heishan, Dapingzi, and Moziping *R. bieti* bands. Fragmentation also leads to reduction in the availability of large food patches, changes in tree species composition and diversity, and introduces edge effects leading to changes in soil moisture and local climate, resulting in a reduction in overall habitat quality (*Laurance, Vasconcelos & Lovejoy, 2000*). Lichen and fruit are important components of the diet of *R. bieti*, accounting for 50% of total feeding time (*Grueter et al., 2009*). Across most forested environments, ripe fruit is characterized by a patchy distribution in space and time. Spatial and temporal fruit patchiness is exacerbated in fragmented landscapes, limiting the ability of species (*Estrada & Coates-Estrada, 1996*). For example, mantled howler monkeys (*Alouatta palliata*) living in more fragmented landscapes spent more time traveling and searching for food than did groups living in less fragmented landscapes (*Estrada & Coates-Estrada, 1996*). We assume that given the large size of *R. bieti* bands, increased fragmentation results in increased challenges in locating sufficient food. Similarly, lichen is known to be negatively impacted by human-induced environmental change such as air pollution, and a decline in the production of lichen is likely to have a severely negative effect on *R. bieti* nutrient intake, reproduction, population growth rates, and survivorship (*Grueter et al., 2008*).

Our results also indicated that the population growth rate of *R. bieti* was negativity correlated with human population density and human accessibility to wildlife habitats. Based on our field observations, we noted that during the *R. bieti* breeding season (month 2 to month 6), local people go into the forest to pick cordyceps, which frightened and affects *R. bieti* breeding behavior by making *R. bieti* spending more time traveling. In addition, human accessibility to wildlife habitats has the potential to become a vector for the transmission of zoonotic diseases between humans, domesticated animals, and wildlife, and rapidly extirpate entire primate subpopulations from local areas (*Patz et al., 2008*; *Lambin et al., 2010*; *Murray & Daszak, 2013*; *Gottdenker et al., 2014*; *Estrada et al., 2018*). We found that in areas with human population densities of greater than 46 human/km$^2$, *R. bieti* growth rates either increased very slowly (rate of 11.4%) or some small bands exhibited negative growth and disappeared from areas such as Bajia, Adong, Houziqin, Heishan, Dapingzi, and Moziping. This reduction has been offset, in part, by the addition of four newly formed bands (Zhina, Gehuaqing, Baijixun (Yongan), Shikuadi) and other bands increasing in size. Moreover, pastures serve as a barrier to migration for *R. bieti* (*Kirkpatrick et al., 1998*; *Grueter et al., 2010*) and therefore severely
limit the movement of individuals across these highly transformed landscapes and the opportunity to expand into unoccupied habitats and form of new bands.

Our results revealed that *R. bieti* populations in the Yunling Nature Reserve and in Jinshichang are the most isolated, and characterized by certain unique haplotypes (haplotype M1-M4, M6-M11, and M29-M30) (*Liu et al., 2007*). Moreover, despite the fact that total population size has increased dramatically over the past 25 years, the population size of three bands (Milaka, Cikatong, and Akou (Anyi)) has markedly decreased, exacerbating the loss of genetic diversity (Table 1). We suggest that management decisions for endangered species should not focus solely on population size but also consider subpopulation genetic diversity. We recommend that effective conservation policies for this species should prioritize protecting certain targeted populations such as Milaka, Cikatong, and Akou (Anyi) that have high intraspecific genetic diversity but low growth rates (Table 1). In addition, government policies that promote the alleviation of human poverty, especially in rural communities, represent an important conservation tool that indirectly protects *R. bieti*. In this regard, the Chinese government has encouraged local farmers to return land to natural forest, prohibited logging, and relocate from Nature Reserves. This serves to reduce human interference and protect habitats that are critical to the survival of *R. bieti* as well as others threatened taxa.

## ACKNOWLEDGEMENTS

We thank the Administration Bureaus of Baimaxueshan National Nature Reserve and Honglaxueshan National Nature Reserve for their support in field work. We also thank Chrissie, Sara, Jenni, and Yu Zhang for their support.

### Funding
This project was supported by the Strategic Priority Research Program of the Chinese Academy of Sciences (XDA19050202), the National Key R & D Program of China (2016YFC0503200), the National Natural Science Foundation of China (31821001) and the State Forestry Administration of China. The funders had no role in study design, data collection and analysis, decision to publish, or preparation of the manuscript.

### Grant Disclosures
The following grant information was disclosed by the authors:
Strategic Priority Research Program of the Chinese Academy of Sciences: XDA19050202.
National Key R & D Program of China: 2016YFC0503200.
National Natural Science Foundation of China: 31821001.
State Forestry Administration of China.

### Competing Interests
The authors declare that they have no competing interests.

## Author Contributions

- Xumao Zhao conceived and designed the experiments, performed the experiments, analyzed the data, contributed reagents/materials/analysis tools, prepared figures and/or tables, authored or reviewed drafts of the paper, approved the final draft.
- Baoping Ren performed the experiments, analyzed the data, contributed reagents/materials/analysis tools.
- Dayong Li performed the experiments, contributed reagents/materials/analysis tools.
- Zuofu Xiang performed the experiments, contributed reagents/materials/analysis tools.
- Paul A. Garber authored or reviewed drafts of the paper, approved the final draft.
- Ming Li conceived and designed the experiments, prepared figures and/or tables, authored or reviewed drafts of the paper, approved the final draft.

## Field Study Permissions

The following information was supplied relating to field study approvals (i.e., approving body and any reference numbers):

This field experiments were approved by State Forestry Administration of China (the Second National Survey on Terrestrial Wildlife Resources in China).

## Data Availability

The raw data are available in Table 1.

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
