# Peer review of "Effects of habitat fragmentation and human disturbance on the population dynamics of the Yunnan snub-nosed monkey from 1994 to 2016"

_PeerJ, doi:10.7717/peerj.6633_

## Round 0.1 · original submission · Major Revisions

Please respond to each of the reviewer comments point-by-point in your rebuttal letter. Pay particular attention to providing more details to describe the methods for the genetic analyses (all three reviewers), and to ensuring a better linkage between the Results and Discussion sections (reviewer 3).

Reviewer 1 ·

Basic reporting

See in general comments

Experimental design

See in general comments

Validity of the findings

See in general comments

Additional comments

This study reports the population size and distribution of the endangered Yunnan snub-nosed monkey, Rhinopithecus bieti, for a period from 1994 to 2016. In this period the population size increased from less than 2000 to about 3000 individuals. According to the authors, this was mainly due to establishing protected habitats. The authors conducted a study that combined census data from different subpopulations in addition to a genetic analysis to determine the abundances and genetic diversity of this metapopulation. Overall, this paper is interesting, well written and relevant, and should be of interest to the readers of PeerJ and other experts in the field.
The one major criticism I have is that the whole aspect of gathering and analyzing genetic information on the 11 subpopulations is grossly neglected. It is unclear whether this was previously published (referred to Liu et al. 2007, 2009) or whether this was new data that was analyzed according to Liu et al. If new information is reported, then this section needs a much more thorough description, including the methods of gathering and analyzing genetic data. If no new data is reported, then this has to be clearly indicated as well as the reasoning why this information was included in this paper.
After reading this paper, the first thought that struck me was that this was a clear example of a successful conservation effort. Clearly there are still concerns about the genetic diversity and increasing human population encroaching on monkey habitats, but it appears that the conservation measures put in place were successful and the population increased by about 50% in 20 years. Surprisingly, there was no real acknowledgement of this success. I don’t think this is any major concern, but I found it a bit curious.
In line with previous comment, I found the first section of the Introduction somewhat dramatic, I don’t think that your general audience of ecologists and conservation biologists need a lot of convincing that biodiversity loss is an issue. I do agree that losing many species is a concern to all of us, but many recent analyses provide a more measured overview of biodiversity loss. Just for scholarly duty, I would recommend for authors to have a look at some of the work by Maria Dornelas or Brian McGill.
Some minor comments:
Ln 210 – negatively instead of negativity
Ln 227-229 – I found it very interesting to learn that lichen is such an important diet item of R. bieti!
Ln 230-235 – As you have not dealt with disease consequences in your study, I think this section is a bit irrelevant. This could be summarized in one sentence, it does not need a whole section.
Figure 2 – in Ln 38 you state that the fragmentation index is between 0 and 1, yet in Figure 2 it goes all the way to 5. Why this discrepancy? Also, based on the shape of your data, I am not convinced that a linear model is the best choice, have you tried fitting different models? Something more similar to an exponentially declining model could fit better and is consistent with other ecological phenomena. Suggest to either transform the data or use another model for a better representation of the decline in growth rates with the fragmentation index.

Reviewer 2 ·

Basic reporting

The reporting of the manuscript is good with some interesting results. However, much more to be done to help improving the clarity of the manuscript. Figures are relevant.

Experimental design

Research questions/objectives should be revised and more clarified with respective methods. The genetics part of the data seems not original and if so, it should be included in the discussion section.

Validity of the findings

The findings are partly good and useful to help conserving Yunnan snub-nosed monkey.

Additional comments

This manuscript aims to evaluate the effects of habitat fragmentation and human disturbance on the population dynamics of the Yunnan snub-nosed monkey is quite good with some interesting results. However, I have the following suggestions for improving the clarity and validity of the manuscript.

Abstract:
1) I recommend authors to restructure the abstract in to four sections as background, methods, results and discussion. The abstract in its form lacks basic information to representing the manuscript.
2) The method section is missing in the abstract section. Delete GLMs from the keywords list.

Introduction:
1) I recommend eliminating the genetics objective, as the authors have no genetic data in the manuscript. However, the authors can use previous research results in the discussion section elsewhere.
2) Line 77: Please replace “Threatened” with “Vulnerable” as “Threatened” stands for Vulnerable, Endangered and Critically Endangered status.
3) The aim of the study in the abstract section (Lines 28-29) and introduction section (Lines 96-98) are conflicting each other. The objective of the study should be clear.
4) The authors should focus on the data they have at hand and previous studies should be included in the discussion section unless the aim of the manuscript is a review paper.

Methods:
1) Line 109: It would be clearer if the authors start a new paragraph here.
2) The authors should clarify the method used for population count. How many time each subpopulation/group counted/censused? Why? Results should be presented in the result section.
3) Line 110 and 111. Please clarify or correct the word “censured”
4) Line 123-125: It seems that the authors have no original genetic data. If so, you could not include in the method section as well as in the results section. However, the authors could use previous studies in the discussion section. I recommend deleting the genetics part from method and result sections.
5) Line 126-151: Landscape and spatial data analysis. I suggest the authors to include “landscape and spatial data analysis” in their objectives to make it clearer.
6) The authors should elaborate how fragment index could be calculated. The definition for “fragment index” should be revised with supportive scientific references.
7) Line 165-167: I suggest deleting the sentence about genetic diversity.
8) Line 176-177: Delete the sentence and discuss this result in the discussion section when necessary.
9) Line 180-181: Delete the sentence .. “Overall, our results …” please also discuss this result in the discussion section.

Discussion:
1) The discussion need restructuring, avoid repetitions and need revision corresponding to your results. It would be great to focus on the interpretation, and implications of your findings. Discuss the effects of habitat fragmentation on the objectives of your study and implications. In the discussion, the authors noted that the effect of habitat fragmentation and degradation causes population decline in the introduction/abstract. However, the population of Yunnan snub-nosed monkey increased in number in some sites and extirpated in other sites. Discuss what it implies clearly.
2) Avoid unnecessary information that belongs to the description of the state-of-the-art of the research topic as they belong to the Introduction. For instance, paragraph 1 of the discussion section.
3) Line 236-246: Please revise this paragraph according to your result.
4) I suggest to include one concluding paragraph at the end of the discussion section.

·

Basic reporting

I have put all comments in the "General comments for the author" section

Experimental design

I have put all comments in the "General comments for the author" section

Validity of the findings

I have put all comments in the "General comments for the author" section

Additional comments

The paper presents a rare case of reassessment of a critically endangered species over a period of 22 years. It should be a good contribution to the study of primate conservation. While the data are highly valuable, the presentation needs some modest level of revision, especially in clarifying some key issues and measurements before it can be published. Additionally, a potential problem as I feel (I may be wrong) that the Discussion section reads rather detached from their results. A tighter connection with the Results section will make the paper stronger and more focused.

The following are the details that I find may need some work. Please pay attention to the issues that I have made substantial comments. Hope they are of some help to their revision.

Lines 35, 135-139: quantifying fragmentation has been attempted many times. (E.g., Kevin R. Crooks 2017. Quantification of habitat fragmentation reveals extinction risk in terrestrial mammals. PNAS.) The authors present a new way of indexing fragmentation. However, they do not provide reasons for doing this. Nor do they provide details for how to do so. I find that in the way as they described (Line 137), I can’t get to the value range of 0-1. A formula may clarify this issue instantaneously. Also, the name “fragment index” is awkward. A better one may be “fragmentation index,” as they presented in Figure 2.

Lines 47-48: Some of the keywords appear to be too general (e.g., “population dynamics, subpopulation growth rates, GLMs; landscape”). Consider deleting them or replacing them with more relevant alternatives. This will increase the probability of hit when readers search for relevant articles.

Line 80: “slow” may be replaced with “long.”

Line 90-91: add a coma after “activities” and replace “has” with “have.”

Line 101-103. The sentence may read better after breaking it into two: change “…Autonomous Region, which is …” to “…Autonomous Region. It is…”

Line 110: change “censured” to “censused” or “surveyed.”

Line 121: Consider simplifying “In order to monitor” to “To monitor.”

Lines 124-125: Please provide information for what exactly was measured and how. This relates to Table 1, in which the authors provided two measurements of genetic diversity. Apparently, they measured the level of heterozygosity but they didn’t provide details about how this was done and why the information was important.

Line 135: strike out the species name “(Abies georgei)” which is not “dark-coniferous forest.”

Line 141: strike out “defined as” – redundant typo. Also, provide the minimal width of pasture land included in the analysis. (2000 meters is only the length.)

Line 153: It may be better to define the meaning of “population” in this study in the Methods section and mark out the names of the populations on the map. This will help readers understand where the populations are.

Line 161: “R. bieti” (missing a dot after “R”)

Line 162: add “and” before “Moziping.”

Lines 179-180: Hard to understand the statistical results as they are presented. Please also provide the p-values. Additionally, it may be better to develop more the meaning of the modeling results in the Discussion section. As it is written, these results are barely mentioned in the Discussion section.

Line 187: “severely” may be replaced with “effectively.”

Line 198: strike out “of” in “collecting of forest products.” Change “domesticated animal grazing within…” to “grazing by domesticated animals within…” Also “domesticated animals” may be replaced by “livestock.”

Line 203: add “and” before “Moziping.” Consider changing “In part this has been…” to “This has been in part …”

Line 237-239: The statement that “it still faces a decrease in genetic diversity …” can be potentially confusing. In principle, when a population is extremely small, the bottleneck effect for the loss of genetic diversity is inevitable with or without habitat fragmentation, the latter of which can only exacerbate (or speed up) the loss of genetic diversity.

---

## Round 0.2 · Minor Revisions

I believe you have done a thorough job in responding to the three reviewers' concerns and suggestions with one exception. With regard to comment 1-2, it seems to me that all three reviewers are asking for further details on exactly how you gathered the genetic data (what kind of samples did you take, from how many individuals, when did you take the samples, etc.). In the revision you have provided information on how the data were analyzed, but not how they were collected. Please provide these additional details so that I can complete the processing of your revised manuscript.

---

## Round 0.3 · accepted · Accept

Thank you for responding to the remaining outstanding comments.

#